# An In-depth Investigation of Sparse Rate Reduction in Transformer-like Models

**Yunzhe Hu**
School of Computing and Data Science
The University of Hong Kong
yzhu@cs.hku.hk

**Difan Zou**
School of Computing and Data Science
& Institute of Data Science
The University of Hong Kong
dzou@cs.hku.hk

**Dong Xu**
School of Computing and Data Science
The University of Hong Kong
dongxu@cs.hku.hk

## Abstract

Deep neural networks have long been criticized for being black-box. To unveil the inner workings of modern neural architectures, a recent work [45] proposed an information-theoretic objective function called Sparse Rate Reduction (SRR) and interpreted its unrolled optimization as a Transformer-like model called Coding Rate Reduction Transformer (CRATE). However, the focus of the study was primarily on the basic implementation, and whether this objective is optimized in practice and its causal relationship to generalization remain elusive. Going beyond this study, we derive different implementations by analyzing layer-wise behaviors of CRATE, both theoretically and empirically. To reveal the predictive power of SRR on generalization, we collect a set of model variants induced by varied implementations and hyperparameters and evaluate SRR as a complexity measure based on its correlation with generalization. Surprisingly, we find out that SRR has a positive correlation coefficient and outperforms other baseline measures, such as path-norm and sharpness-based ones. Furthermore, we show that generalization can be improved using SRR as regularization on benchmark image classification datasets. We hope this paper can shed light on leveraging SRR to design principled models and study their generalization ability.

## 1 Introduction

Transformers [39, 11] have become the de facto choice of neural architecture nowadays and find great success in applications across language, vision, speech, and other scientific fields. The self-attention module in Transformers utilize global interactions to capture long-range dependency. However, the mechanisms and learning process of self-attention and other components in Transformers remain open problems, calling for more research to interpret and understand their properties.

One approach to interpreting the attention module involves experimental observation of the attention module to gain insights into their behaviors. For instance, DINO [7] provides a means to observe and analyze attention maps w.r.t class tokens in Vision Transformer (ViT), shedding light on their emerging interpretability from self-supervised learning. Another line of work focuses on interpreting or building attention module and even Transformer-like models from a mathematical perspective. Works in this vein have attempted to establish connections between Transformers and a reverse-engineered energy

38th Conference on Neural Information Processing Systems (NeurIPS 2024).

function [43, 17], associative memory such as modern Hopfield network [35, 37, 26, 4] and sparse distributed memory [5], or programming languages [41, 22], to name a few.

Recently, the study of algorithm unrolling has emerged as a promising technique to bridge the gap between iterative optimization and neural architecture. A work by Yu et al. [45] considers the objective of representation learning as optimizing the Sparse Rate Reduction (SRR), a function that promotes maximum information gain described by the coding rate function [24, 46] and induces sparsity. In particular, they show that a Multi-head Subspace Self-Attention (MSSA) operator with skip connection and an Iterative Shrinkage-Thresholding Algorithms (ISTA) operator can be derived under some assumptions by unrolling minimization of the coding rate of representations in incoherent subspaces, i.e., compression and sparse coding, i.e., sparsification, respectively. By stacking these operations into layers, they build a Transformer-like model CRATE in which every layer should have the completely interpretable *compress-then-sparsify* behavior. However, although motivated by an information-theoretic and principled objective, it is still unexplored whether the core component MSSA operator with skip connection indeed implements the idea of compression in practice and how information propagates in the forward pass. On the other hand, SRR as the objective of representation learning is still an empirical formulation. Its causal relationship to generalization remains elusive.

In this paper, we conduct an in-depth investigation of this Transformer-like model and take steps to address these limitations. Our contributions are summarized as follows:

- In Section 4, we highlight the derivation artifacts through analysis of the key component MSSA operator and explore implementation variants of CRATE by inspecting the layer-wise behaviors. We show that the gradient approximation of the compression term will yield a counterproductive effect, performing *decompression* of token representations instead.

- In Section 5, we uncover the correlation between the learning objective SRR and generalization in unrolled models. By training models with varied hyperparameters, we show that SRR as a complexity measure has a positive correlation coefficient and outperforms other baselines.

- In Section 6, we demonstrate the effectiveness of SRR as a regularization technique for improved performance on benchmark datasets. Specifically, we show that the classification accuracy of unrolled models on CIFAR-10/100 can be consistently improved using a simple and efficient implementation of regularization.

## 2 Related Work

### 2.1 Interpreting Transformers

Research on interpreting Transformers [39, 11] has surged recently. Despite its achievements, the mechanisms and learning of attention layers remain enigmatic. One approach to interpreting Transformers is to experimentally observe the inner representations or output of key components like self-attention. This includes analysis by projecting parameters of Transformers to embedding space [10], inspecting the representations with another language model [14], visualizing attention map [7, 8, 44], etc. Several works opt for "mechanistic interpretability" [12, 28, 40] aiming to reverse-engineer the representations learned by Transformers that have "grokked" or mastered complex modular arithmetic task [34] and other synthetic tasks [23, 47]. Another line of work focuses more on theoretical understanding and building connections to other concepts. These papers utilize tools such as Bayesian inference [1], convex optimization [36] to analyze attention in Transformers. There have also been attempts to interpret a Transformer as an energy function optimizer [43, 17], connect attention to memory [35, 37, 26, 4, 5] or interacting particle systems [13] or transform into human-readable programs [41, 22], to name just a few. Our work focuses on the empirical investigation of a Transformer-like model, CRATE [45], recently introduced from pure mathematical derivation.

### 2.2 Algorithm Unrolling

Algorithm unrolling [27] has emerged as a promising technique for designing interpretable and efficient deep learning architectures. This approach establishes a direct connection between iterative algorithms and neural architecture, with each iteration of the algorithm corresponding to one layer of the architecture. Previous works have employed this technique to design popular networks in a forward-constructed manner. For instance, the seminal work [15] proposed to unroll the Iterative

Shrinkage-Thresholding Algorithm for sparse coding into layers of linear operation followed by ReLU non-linearity. Other works have tried to find a representation objective function to unroll into convolutional neural network [33, 6], graph neural network [42], and Transformers [43, 17]. We will follow this iteration-layer correspondence to conduct layer-wise analysis.

## 3   Revisiting Sparse Rate Reduction

Let $\boldsymbol{Z} = [\boldsymbol{z}_1, \ldots, \boldsymbol{z}_N] \in \mathbb{R}^{d \times N}$ denote $N$ samples, where each column $\boldsymbol{z}_i \in \mathbb{R}^d$ represents tokens in Transformers. $\boldsymbol{U} = [\boldsymbol{U}_1, \ldots, \boldsymbol{U}_K] \in \mathbb{R}^{d \times Kp}$ denote a set of incoherent basis spanning $K$ subspaces, wherein columns of $\boldsymbol{U}_i \in \mathbb{R}^{d \times p}$ represent basis in $i$-th low-dimensional subspace ($p < d$). We follow the configuration that $d = Kp$ as in standard ViT [11].

Previously, Yu et al. [46] propose that the compactness of representations $\boldsymbol{Z} \in \mathbb{R}^{d \times N}$ can be measured by a coding rate function: $R(\boldsymbol{Z}) \doteq \frac{1}{2} \log \det(\boldsymbol{I} + \frac{d}{N\epsilon^2} \boldsymbol{Z}^T \boldsymbol{Z})$. A more recent study [45] contends that the objective of representation learning is to transform and compress samples from an unknown distribution to a mixture of low-dimensional Gaussian distributions supported on incoherent bases. This objective boils down to the maximization of *Sparse Rate Reduction* (SRR):

$$\max_{\boldsymbol{Z} \in \mathbb{R}^{d \times N}} R(\boldsymbol{Z}) - R^c(\boldsymbol{Z}; \boldsymbol{U}) - \lambda \|\boldsymbol{Z}\|_0, \tag{1}$$

where $\| \cdot \|_0$ means $\ell_0$ norm and $R^c(\boldsymbol{Z}; \boldsymbol{U}) \doteq \sum_{k=1}^{K} R(\boldsymbol{U}_k^T \boldsymbol{Z})$ measures the compactness of representations in the low-dimensional subspaces. One layer of a network, formulated as a mapping $f_{\boldsymbol{w}}(\cdot)$ parameterized by $\boldsymbol{w}$, can be interpreted as applying one step of gradient-based methods to the objective in (1). In practice, Yu et al. [45] use alternating minimization to break down the optimization into two steps: *compression*, i.e. $\min_{\boldsymbol{Z}} R^c(\boldsymbol{Z}; \boldsymbol{U})$ and *Sparsification*, i.e. $\min_{\boldsymbol{Z}} \lambda \|\boldsymbol{Z}\|_0 - R(\boldsymbol{Z})$. Specifically, given representation $\boldsymbol{Z}^{\ell-1}$ at $(\ell - 1)$-th layer, $\boldsymbol{Z}^\ell$ can be obtained by two-step optimization:

$$\boldsymbol{Y}^\ell = \boldsymbol{Z}^{\ell-1} - \alpha \nabla R^c(\boldsymbol{Z}^{\ell-1}; \boldsymbol{U}^\ell) \approx \boldsymbol{Z}^{\ell-1} + \alpha \gamma^2 \, \mathrm{MSSA}(\boldsymbol{Z}^{\ell-1}; \boldsymbol{U}^\ell), \tag{2}$$

$$\boldsymbol{Z}^\ell = \mathrm{ReLU}\left(\boldsymbol{Y}^\ell + \beta(\boldsymbol{D}^\ell)^T(\boldsymbol{Y}^\ell - \boldsymbol{D}^\ell \boldsymbol{Y}^\ell) - \beta\lambda\mathbf{1}\right), \tag{3}$$

where $\alpha, \beta > 0$ are step sizes, $\boldsymbol{D}^\ell \in \mathbb{R}^{d \times d}$ is assumed as a complete dictionary, scalar $\gamma \doteq \frac{p}{N\epsilon^2}$ and

$$\mathrm{MSSA}(\boldsymbol{Z}; \boldsymbol{U}) = \sum_{k=1}^{K} \boldsymbol{U}_k \boldsymbol{U}_k^T \boldsymbol{Z} \, \mathrm{softmax}((\boldsymbol{U}_k^T \boldsymbol{Z})^T (\boldsymbol{U}_k^T \boldsymbol{Z}))$$

$$= [\boldsymbol{U}_1, \ldots, \boldsymbol{U}_K] \begin{bmatrix} \boldsymbol{U}_1^T \boldsymbol{Z} \, \mathrm{softmax}((\boldsymbol{U}_1^T \boldsymbol{Z})^T (\boldsymbol{U}_1^T \boldsymbol{Z})) \\ \vdots \\ \boldsymbol{U}_K^T \boldsymbol{Z} \, \mathrm{softmax}((\boldsymbol{U}_K^T \boldsymbol{Z})^T (\boldsymbol{U}_K^T \boldsymbol{Z})) \end{bmatrix} \tag{4}$$

The operator $\mathrm{MSSA}(\cdot; \boldsymbol{U})$ in (4), called the Multi-head Subspace Self-Attention (MSSA) operator, takes the form of self-attention in standard Transformers [39, 11], with tied query, key and value matrix, i.e., $\boldsymbol{U}_k^T$ while the output matrix being its transpose, i.e. $\boldsymbol{U}_k$. Instead of strictly following this formulation, they further replace $[\boldsymbol{U}_1, \ldots, \boldsymbol{U}_K] \in \mathbb{R}^{d \times Kp}$ in the MSSA operator with an additional learnable parameter $\boldsymbol{W} \in \mathbb{R}^{d \times Kp}$. To distinguish them, we name the model with implementation (4) **CRATE-C(onceptual)**. By incrementally optimizing (1) with alternating minimization, a Transformer-like model with layered structures can be naturally constructed. With input $\boldsymbol{Z}^0$, e.g. tokenized images in ViT, an $L$-layer model iteratively optimizes the input and yields the final representations $\boldsymbol{Z}^L$. Parameters $\{\boldsymbol{U}^\ell\}_{\ell=1}^L$ and $\{\boldsymbol{D}^\ell\}_{\ell=1}^L$ can be learned through end-to-end training [15].

## 4   Is Sparse Rate Reduction Optimized in Transformer-like Models?

While the white-box Transformer-like model proposed in [45] is derived by unrolling optimization upon a pre-defined objective function, whether the optimization is implemented by the model in the forward pass is still unclear. In this section, we first review the main derivations at the core of building CRATE, i.e. unrolling optimization $\min_{\boldsymbol{Z}} R^c(\boldsymbol{Z}; \boldsymbol{U})$ into MSSA operator with skip connection as in (2), and identify the pitfalls in implementing the minimization. We then provide variant models based on different implementations and empirically show their layer-wise behaviors.

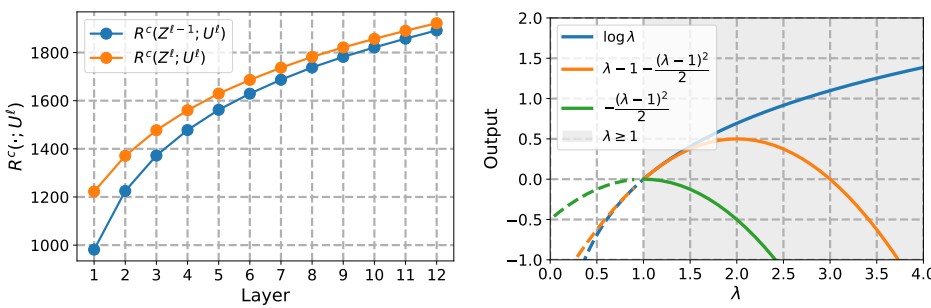

(a) Changes of coding rate projected on the same set of basis at each layer.

(b) Different approximation of $\log(\cdot)$.

Figure 1: In a simplified attention-only experiment, MSSA operator with skip connection actually implements an ascent method on $R^c(\boldsymbol{Z}; \boldsymbol{U})$, opposed to its design purpose (*left*). This is due to an artifact in approximation with its second-order term. (*right*)

## 4.1 Pitfalls in Deriving CRATE-C

We first show that the second-order Taylor expansion of the coding rate of representations $\boldsymbol{Z}$ projected onto subspaces can be expressed as:

$$
R^c(\boldsymbol{Z}; \boldsymbol{U}) = \sum_{k=1}^{K} \sum_{i=1}^{N} \frac{1}{2} \log \lambda_i^k \geq \sum_{k=1}^{K} \sum_{i=1}^{N} \frac{1}{2} \left( \lambda_i^k - 1 - \frac{(\lambda_i^k - 1)^2}{2} \right)
$$
$$
= \sum_{k=1}^{K} \left( \underbrace{\frac{\gamma}{2} \|\boldsymbol{U}_k^T \boldsymbol{Z}\|_F^2}_{\text{First-order term}} \underbrace{- \frac{\gamma^2}{4} \|(\boldsymbol{U}_k^T \boldsymbol{Z})^T \boldsymbol{U}_k^T \boldsymbol{Z}\|_F^2}_{\text{Second-order term}} \right), \tag{5}
$$

where $\lambda_i^k \geq 1, i \in [N]$ are the eigenvalues of $\boldsymbol{I} + \gamma (\boldsymbol{U}_k^T \boldsymbol{Z})^T \boldsymbol{U}_k^T \boldsymbol{Z}$. Following the derivation and implementation from Appendix A.2 in [45], the MSSA operator with skip connection is constructed by performing an *approximation* of gradient descent on $R^c(\boldsymbol{Z}; \boldsymbol{U})$:

$$
\boldsymbol{Z} - \alpha \nabla_{\boldsymbol{Z}} R^c(\boldsymbol{Z}; \boldsymbol{U}) = \boldsymbol{Z} - \alpha \gamma \sum_{k=1}^{K} \boldsymbol{U}_k \boldsymbol{U}_k^T \boldsymbol{Z} \left( \boldsymbol{I} + \gamma (\boldsymbol{U}_k^T \boldsymbol{Z})^T (\boldsymbol{U}_k^T \boldsymbol{Z}) \right)^{-1} \tag{6}
$$

$$
\approx \boldsymbol{Z} - \alpha \left( \underbrace{\gamma \sum_{k=1}^{K} \boldsymbol{U}_k \boldsymbol{U}_k^T \boldsymbol{Z}}_{\nabla \text{ of first-order term}} \underbrace{- \gamma^2 \sum_{k=1}^{K} \boldsymbol{U}_k \boldsymbol{U}_k^T \boldsymbol{Z} (\boldsymbol{U}_k^T \boldsymbol{Z})^T (\boldsymbol{U}_k^T \boldsymbol{Z})}_{\nabla \text{ of second-order term}} \right) \tag{7}
$$

$$
\approx \boldsymbol{Z} + \alpha \gamma^2 \sum_{k=1}^{K} \boldsymbol{U}_k \boldsymbol{U}_k^T \boldsymbol{Z} \, \text{softmax}((\boldsymbol{U}_k^T \boldsymbol{Z})^T (\boldsymbol{U}_k^T \boldsymbol{Z})). \tag{8}
$$

It can be seen that this update step takes the gradient of a lower bound of $R^c(\boldsymbol{Z}; \boldsymbol{U})$ and discards the first-order term. With a proper step size, the coding rate on the same subspaces is expected to decrease after one iteration. However, we will show that this is not the actual case via a toy experiment.

We consider a simplified setting where $L$ layers of update (8) are conducted with parameters $\{\boldsymbol{U}^\ell\}_{\ell=1}^L$ initialized as orthonormal matrices. We initialize a random variable $\boldsymbol{Z}^0$ from a Gaussian distribution and measure the coding rate before and after each layer. We set $N = 196$, $L = 12$, $d = 384$, $K = 6$, $\alpha = 1$, and a proper $\epsilon^2$ such that $\gamma = 1$. As shown in Figure 1a, $R^c(\boldsymbol{Z}^\ell; \boldsymbol{U}^\ell)$ is always greater than $R^c(\boldsymbol{Z}^{\ell-1}; \boldsymbol{U}^\ell)$ and $R^c(\boldsymbol{Z}^\ell; \boldsymbol{U}^\ell)$ is increasing in general as the layer goes deeper. This means the update (8) that resembles the standard self-attention with skip connection does not essentially implement a descent method on $R^c$. The crux lies in the approximation of $R^c$'s gradient.

When taking the gradient of $R^c$ to construct the MSSA operator, omitting its first-order term will produce a counterproductive effect. As shown on the left-hand side of the inequality in (5), $R^c$ can be

expressed as the sum of logarithms of eigenvalues. We expect the eigenvalues to decrease to minimize the value of $R^c$. Figure 1b illustrates different approximations of the logarithm function. If we omit the first-order term of its Taylor expansion and only perform descent methods on its second-order term (corresponding to $-\frac{(\lambda_i^k - 1)^2}{2}$), the eigenvalues will go up leading to an increase in the value of $R^c$. Therefore, one step of update (8) secretly maximizes $R^c$, contrary to the purpose of its design. More figures detailing this issue are in Appendix A.

## 4.2 Producing CRATE Variants

In the previous subsection, we show the problems arising from gradient approximation when unrolling $\min_{\boldsymbol{Z}} R^c(\boldsymbol{Z}; \boldsymbol{U})$ into MSSA operator with shortcut. We will, in the subsection, introduce two variants of CRATE induced by the conceptual and implementation gaps. These variants can be considered as the alternative instantiations of the optimization-induced architectures but in a more self-contained way. They also serve as representative samples for our subsequent investigations of SRR.

One variant of CRATE, motivated by the theoretical gap between CRATE-C and the SRR principle, could naturally emerge when the sign before the MSSA operator in (8) is changed. Similar to previous analysis via eigenvalues, this update of representations in fact implements one step of ascent methods on the second-order term of $\boldsymbol{R}^c$, therefore minimizing the eigenvalues and consequently $\boldsymbol{R}^c$. This variant is designed to counter the pitfalls in CRATE-C, enabling a more faithful reduction in $\boldsymbol{R}^c$ and thereby enhancing alignment with the SRR principle. We term the Transformer-like model with this implementation **CRATE-N(egative)**:

$$\boldsymbol{Z} - \alpha\gamma^2 \sum_{k=1}^{K} \boldsymbol{U}_k \boldsymbol{U}_k^T \boldsymbol{Z} \operatorname{softmax}((\boldsymbol{U}_k^T \boldsymbol{Z})^T (\boldsymbol{U}_k^T \boldsymbol{Z})). \tag{9}$$

The other variant we would like to introduce is motivated by the misalignment between CRATE and CRATE-C. Although replacing the output matrix $[\boldsymbol{U}_1, \ldots, \boldsymbol{U}_K]$ with learnable parameters $\boldsymbol{W}$ in CRATE empirically boosts performance, it also contaminates the framework and sacrifices the mathematical interpretability. Does this modification really matter? Can we preserve model performance while maintaining framework integrity? It turns out that a simple transpose operation of the output matrix could greatly close the empirical gap to CRATE, without more parameters. Other manipulations and discussions can be found in Appendix B. We refer to the model with this simple manipulation **CRATE-T(ranspose)**:

$$\boldsymbol{Z} + \alpha\gamma^2 \left[\boldsymbol{U}_1, \ldots, \boldsymbol{U}_K\right]^T \begin{bmatrix} \boldsymbol{U}_1^T \boldsymbol{Z} \operatorname{softmax}((\boldsymbol{U}_1^T \boldsymbol{Z})^T (\boldsymbol{U}_1^T \boldsymbol{Z})) \\ \vdots \\ \boldsymbol{U}_K^T \boldsymbol{Z} \operatorname{softmax}((\boldsymbol{U}_K^T \boldsymbol{Z})^T (\boldsymbol{U}_K^T \boldsymbol{Z})) \end{bmatrix}. \tag{10}$$

## 4.3 Behaviors of Sparse Rate Reduction

The Transformer-like model CRATE is built by sequentially stacking the layer that comprises two modules in (2) and (3) (or different implementations). Although each module is designed to implement one-step optimization of different objectives, it is unclear whether the architecture design achieves the optimization as a whole. On the other hand, there is also a need to determine whether the model parameters learned through end-to-end training actually lead to improved optimization.

To investigate how sparse rate reduction evolves in the forward pass and during training, we train CRATE and its variants on CIFAR-10/100 datasets and evaluate the sparse rate reduction measure $\lambda \|\boldsymbol{Z}^\ell\|_0 + R^c(\boldsymbol{Z}^\ell; \boldsymbol{U}^\ell) - R(\boldsymbol{Z}^\ell)$ at different layers and epochs on the training set. $\lambda$ is chosen as 0.1 and detailed experiment settings can be found in Section 6.1. Figure 2 and Figure 3 show the behaviors of sparse rate reduction of CRATE along with its variants CRATE-C, CRATE-N, and CRATE-T under Tiny configurations in [45]. When the models are randomly initialized, the sparse rate reduction measure almost monotonically decreases in the first 9 layers and then rises in the subsequent layers. This partly confirms the layer-wise optimization of the objective SRR and its alignment with forward architecture design, although in Section 4.1 we demonstrate that $R^c(\boldsymbol{Z}; \boldsymbol{U})$ will monotonically go up in the absence of operation (3). We conjecture that the ReLU non-linearity may also play an important role in optimizing the compression term $R^c(\boldsymbol{Z}; \boldsymbol{U})$ in the forward pass.

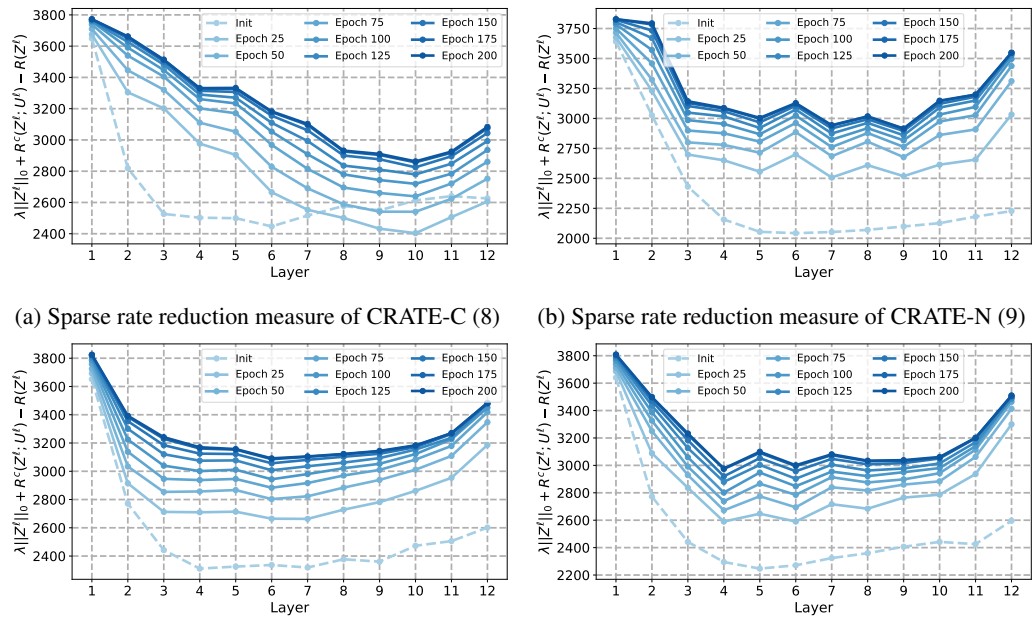

(a) Sparse rate reduction measure of CRATE-C (8)

(b) Sparse rate reduction measure of CRATE-N (9)

(c) Sparse rate reduction measure of CRATE-T (10)

(d) Sparse rate reduction measure of CRATE

Figure 2: Sparse rate reduction measure $\lambda\|\boldsymbol{Z}\|_0 + R^c(\boldsymbol{Z}; \boldsymbol{U}) - R(\boldsymbol{Z})$ of CRATE and its variants evaluated at different layers and epochs on CIFAR-10.

Another surprising finding is that as the learning process proceeds, the sparse rate reduction measure at each layer will increase monotonically across all models, with a rare exception in the last few layers of CRATE-C.

These phenomena give us implications for understanding Transformer-like models: the representations of initialized models converge fast in the first few layers and hover around the local minimum of the objective landscape; however, the useful information in representations may be discarded due to over-compression and the learning of parameters gradually increases sparse rate reduction measure to counteract this effect for improved task-specific representations.

To summarize, our finding is that sparse rate reduction measure is incrementally optimized in a realistic setting at initialization. This aligns well with its design purpose from a macro perspective. With varied implementations, the result still holds even when the compression-inspired operator MSSA diverges from its goal from a micro perspective. We postulate that ReLU non-linearity in (3) could also promote compression and leave their interaction for future work.

## 5 Whether Sparse Rate Reduction Benefits Generalization?

So far, we have partially confirmed the validity of different implementations of Transformer-like models by inspecting the layer-wise optimization of SRR. But whether this objective is important or principled for these architectures to generalize is still an unaddressed problem. In this section, we want to explore the predictive power of SRR and its causal relationship to the generalization of CRATE.

### 5.1 Sparse Rate Reduction as a Complexity Measure

An important tool to study the generalization of deep networks is *complexity measure*. A complexity measure that can properly reflect the generalization needs to have the following property: lower complexity should indicate a smaller generalization gap. Complexity measures can be either theoretically motivated, such as PAC-Bayes [25, 29], VC-dimension [38], norm-based bounds [32, 3, 30] or empirically motivated, such as sharpness [20] and path-norm [31]. We choose to adapt SRR into a

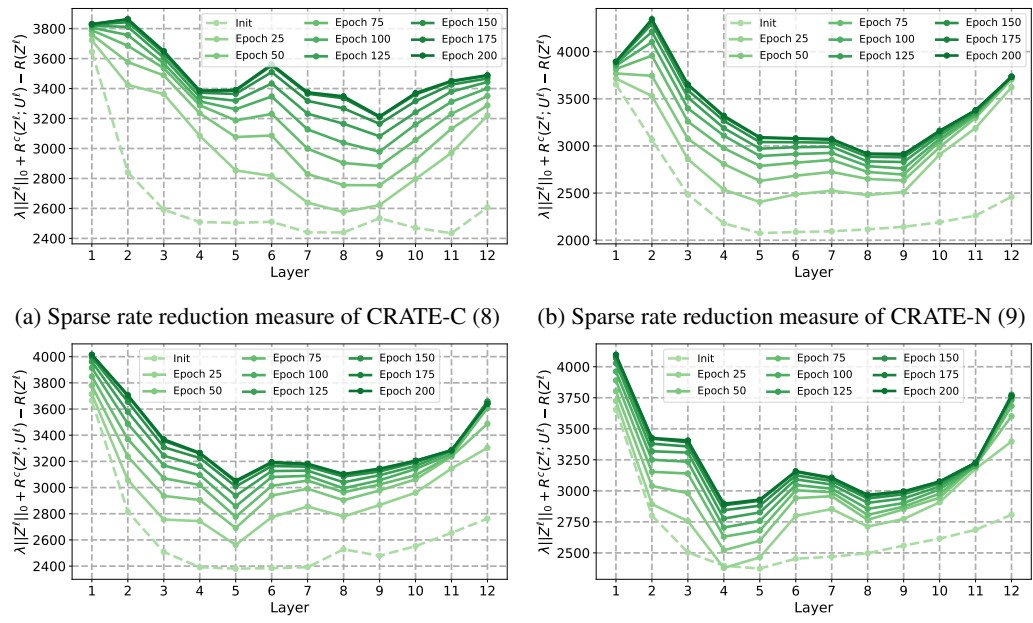

(a) Sparse rate reduction measure of CRATE-C (8)  (b) Sparse rate reduction measure of CRATE-N (9)

(c) Sparse rate reduction measure of CRATE-T (10)  (d) Sparse rate reduction measure of CRATE

Figure 3: Sparse rate reduction measure $\lambda\|\boldsymbol{Z}\|_0 + R^c(\boldsymbol{Z};\boldsymbol{U}) - R(\boldsymbol{Z})$ of CRATE and its variants evaluated at different layers and epochs on CIFAR-100.

complexity measure that belongs to the latter category:

$$\mu_{\mathrm{SRR}}(\boldsymbol{w};\boldsymbol{Z}) = \frac{1}{L}\sum_{\ell=1}^{L}\mu_{\mathrm{SRR}}^{\ell}(\boldsymbol{w}^{\ell};\boldsymbol{Z}^{\ell}) = \frac{1}{L}\sum_{\ell=1}^{L}\left(\lambda\|\boldsymbol{Z}^{\ell}\|_0 + R^c(\boldsymbol{Z}^{\ell};\boldsymbol{U}^{\ell}) - R(\boldsymbol{Z}^{\ell})\right), \qquad (11)$$

where $\boldsymbol{Z}^{\ell}$ denotes the output at layer $\ell$ and $\boldsymbol{w}^{\ell}$ contains the parameters including $\boldsymbol{U}^{\ell}$ and $\boldsymbol{D}^{\ell}$.

## 5.2 Correlation with Generalization

An effective measure of complexity should bound the generalization gap, defined as the difference between validation loss and training loss when the latter reaches a threshold, i.e., $\mathcal{L}_{val} - \mathcal{L}_{train}$, with high probability. However, for those measures that do not provably bound this gap, as is the case with SRR measure (11), we need to evaluate its correlation with the generalization gap to understand its causal relationship to generalization.

**Collecting Trained Models**  To evaluate the complexity measure and generalization across models, we consider changing the hyperparameters and collect a set of models trained to meet a specific stopping criterion. Here, we also consider the model type containing different variants of CRATE as a hyperparameter to investigate its influence on generalization. Formally, let $\Theta_i$ denote a type of hyperparameter with $|\Theta_i|$ different choices, and define $\boldsymbol{\theta} \doteq (\theta_1, \theta_2, \ldots, \theta_n) \in \Theta_1 \times \cdots \times \Theta_n$ as an instantiation from $n$ types of hyperparameters. By varying choices across hyperparameter space, we can produce $|\Theta_1| \times \cdots \times |\Theta_n|$ models. In our experiment, we consider $n = 5$ hyperparameters, including *batch size*, *initial learning rate*, *width*, *dropout*, and *model type*. Each contains 2 choices except that the model type contains 4 implementations we discussed before. We successfully train a total of 64 models on CIFAR-10 dataset, when cross-entropy loss reaches 0.01 following the stopping criterion in [18]. Experimental details and choices of hyperparameters can be found in Appendix C.

**Evaluation Criterion**  A common method for measuring correlation is by utilizing Kendall's rank correlation coefficient [19, 18], which ranges from -1 to 1. Generally, the closer the coefficient is to one, the stronger the causal relationship and the greater the predictive power a measure can offer for generalization. Zero value usually means independent relationships. For a given complexity

Table 1: Correlation of complexity measures with generalization gap (width $d = 384$).

| Complexity measures | Batch size | Learning rate | Dropout | Model type | Overall $\tau$ | $\Psi$ |
|---|---|---|---|---|---|---|
| $\ell_2$-norm | 0.200 | -0.333 | -0.333 | -0.429 | -0.363 | -0.224 |
| $\ell_2$-norm-init | 0.200 | -0.200 | -0.333 | -0.286 | -0.290 | -0.158 |
| # params | 0.000 | 0.000 | 0.000 | -0.572 | -0.351 | -0.143 |
| 1/margin | -0.067 | 0.467 | 0.467 | 0.238 | 0.415 | 0.276 |
| sum-of-spec | 0.200 | -0.333 | -0.467 | -0.381 | -0.290 | -0.245 |
| prod-of-spec | 0.200 | -0.333 | -0.467 | -0.476 | -0.338 | -0.269 |
| sum-of-spec/margin | 0.333 | -0.333 | -0.467 | -0.048 | -0.230 | -0.129 |
| prod-of-spec/margin | 0.333 | -0.333 | -0.467 | -0.143 | -0.260 | -0.152 |
| fro/spec | -0.200 | 0.333 | 0.467 | -0.476 | 0.019 | 0.031 |
| spec-init-main | 0.333 | -0.333 | -0.467 | -0.190 | -0.273 | -0.164 |
| spec-orig-main | 0.200 | -0.333 | -0.467 | -0.095 | -0.252 | -0.174 |
| sum-of-fro | 0.200 | -0.333 | -0.333 | -0.381 | -0.325 | -0.212 |
| prod-of-fro | 0.200 | -0.333 | -0.333 | -0.429 | -0.372 | -0.224 |
| sum-of-fro/margin | 0.333 | -0.200 | -0.467 | -0.048 | -0.217 | -0.095 |
| prod-of-fro/margin | 0.333 | -0.200 | -0.467 | -0.143 | -0.247 | -0.119 |
| fro-distance | 0.200 | -0.200 | -0.333 | -0.286 | -0.290 | -0.155 |
| spec-distance | 0.200 | -0.200 | -0.333 | -0.286 | -0.290 | -0.155 |
| param-norm | 0.200 | -0.333 | -0.333 | -0.429 | -0.363 | -0.224 |
| path-norm | 0.333 | -0.600 | -0.467 | -0.286 | -0.191 | -0.255 |
| pac-bayes-init | 0.200 | 0.200 | -0.600 | 0.238 | 0.015 | -0.009 |
| pac-bayes-orig | -0.200 | 0.333 | 0.467 | 0.381 | 0.333 | 0.245 |
| $1/\sigma$ pac-bayes-flatness | -0.267 | 0.333 | 0.333 | 0.455 | 0.333 | 0.213 |
| SRR | -0.067 | 0.467 | 0.333 | 0.714 | **0.445** | **0.362** |

measure, we can construct a set of samples $\mathcal{T}$ containing the measure $\mu(\boldsymbol{\theta})$ and generalization gap $g(\boldsymbol{\theta})$ evaluated at different combinations of hyperparameters $\boldsymbol{\theta}$ and calculated Kendall's coefficient on this set:

$$\mathcal{T} \triangleq \cup_{\boldsymbol{\theta} \in \Theta_1 \times \cdots \times \Theta_n} \{(\mu(\boldsymbol{\theta}), g(\boldsymbol{\theta}))\}, \tag{12}$$

$$\tau(\mathcal{T}) \triangleq \frac{1}{|\mathcal{T}|(|\mathcal{T}| - 1)} \sum_{(\mu_1, g_1) \in \mathcal{T}} \sum_{(\mu_2, g_2) \in \mathcal{T} \setminus (\mu_1, g_1)} \text{sign}(\mu_1 - \mu_2) \text{sign}(g_1 - g_2). \tag{13}$$

**Experimental Results.** In our experiment, we find that the correlation of various measures with the generation can be reflected with more prominence under a selected width. Accordingly, we present the results in terms of Kendall's coefficient $\tau$ in Table 1 and scatter plot of SRR measure in Figure 4 when the width $d$ is chosen as 384. The results when $d = 768$ is deferred to Appendix D. The granulated coefficient $\Psi$ is also reported (see [18] for a detailed definition).

We confirm the findings from prior works that some norm-based measures, such as sum/prod of spectral/Frobenius norm of parameters negatively correlate with generalization, even on Transformer-like models. An interesting finding is that path-norm also negatively correlates with generalization, which partly contradicts the previous conclusion. This implies that regularization on path-norm, e.g. Path-SGD [31], may not be applicable for improved generalization on Transformer-like models. Among the measures we investigated, the inverse of margin and sharpness-based PAC-Bayes flatness show positive and strong correlations. This result justifies the common belief that larger margin or flatter loss landscape leads to better generalization across the investigated Transformer-like models. Compared to baselines, the SRR measure in (11)

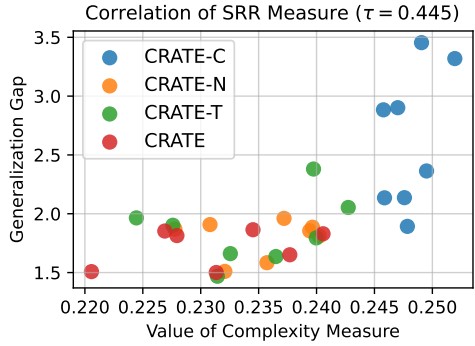

Figure 4: A scatter plot illustrating the value of SRR measure and generalization gap across CRATE and variants with network width $d = 384$.

achieves the highest overall coefficient and, particularly in the model type axis, outperforms the rest. This motivates the use of SRR as regularization in the loss function to improve generalization.

# 6 Sparse Rate Reduction as Regularization

Since SRR measure enjoys a strong correlation to the generalization of Transformer-like models, we would like to investigate its potential as the direct regularization to the standard training loss. In particular, we add the SRR measure in (11) by a regularization factor $\eta$ to the cross-entropy loss:

$$\min_{\boldsymbol{w}} \mathcal{L}_{\text{ce}}(\boldsymbol{w}) + \lambda \cdot \frac{1}{L} \sum_{\ell=1}^{L} \mu_{\text{SRR}}^{\ell}(\boldsymbol{w}^{\ell}; \boldsymbol{Z}_{\text{StopGrad}}^{\ell}), \tag{14}$$

where $\lambda > 0$ is the regularization coefficient and $\boldsymbol{Z}_{\text{StopGrad}}^{\ell} = f_{\boldsymbol{w}^{\ell}}(\text{StopGrad}(\boldsymbol{Z}^{\ell-1}))$. The operator $\text{StopGrad}$ here, implemented as "$\text{Tensor.detach}()$" in PyTorch, prevents gradient propagation from the output $\boldsymbol{Z}^{\ell}$ to the previous layers. This allows parameters $\boldsymbol{w}^{\ell}$ at each layer to be updated without interfering with each other, giving more precise optimization of SRR in separate layers.

## 6.1 Experiment Settings

**Model Configurations**   We follow the configuration of CRATE-Tiny in [45] in this experiment. Specifically, we set the depth $L = 12$, width $d = 384$, number of subspaces $K = 6$, step size $\alpha = 1$, and scaling factor $\gamma = 1$. We also include LayerNorm before each operation in (2)(3) for better trainability and learnable positional encoding. A trainable [CLS] token is prepended to the representations for computing cross-entropy loss and classification.

**Datasets and Optimization**   We use CIFAR-10 and CIFAR-100 datasets for training and evaluation. In practice, we adopt Adam [21] optimizer and initialize learning rate as $1 \times 10^{-4}$ with cosine decay. All models are trained for 200 epochs with batch size as 128. Note that we only use the basic data augmentations: random resize and cropping, horizontal flipping, and RandAugment [9] (with the number transformations $n = 2$ and magnitude $m = 14$). We do not use other techniques for state-of-the-art performance but to demonstrate the effectiveness of SRR as regularization. We tune the factor $\eta$ via a grid search over $\{0.0001, 0.001, 0.01, 0.1, 1\}$ and find that 0.001 works best. All experiments are conducted on NVIDIA GeForce RTX 3090.

Table 2: Top-1 accuracy for CRATE and its variants trained with or without SRR regularization on CIFAR-10/100 from scratch (width $d = 384$).

| Models | CIFAR-10 | | CIFAR-100 | |
|---|---|---|---|---|
| | cross-entropy | + SRR regularization (L=12) | cross-entropy | + SRR regularization (L=12) |
| CRATE-C | 76.87 | **77.61** | 43.40 | **44.53** |
| CRATE-N | 81.52 | **81.91** | 55.11 | **55.62** |
| CRATE-T | 85.49 | **85.52** | 60.59 | **60.69** |
| CRATE | 86.67 | **86.79** | 62.40 | **62.52** |

## 6.2 Efficient Implementation

Regularizing the training loss with sparse rate reduction measure (11) needs to compute $R(\boldsymbol{Z})$ and $R^c(\boldsymbol{Z}; \boldsymbol{U})$ for every layer. However, this is highly inefficient as it involves high-dimensional matrix multiplication, and it lacks flexibility in controlling parameters. To alleviate this issue, we implement efficient regularization as per layer regularization or random layer regularization: select a pre-defined layer or a random layer with uniform probability during training. In practice, we find that the former works better. Table 2 provides the results of CRATE and its variants trained from scratch on CIFAR-10/100. SRR regularization is sufficient to improve the performance by simply leveraging the last layer. We also provide a comparison of efficient implementations in Appendix E

# 7   Conclusion

To further research in interpreting neural architecture, we provide an in-depth investigation of a recent mathematically driven Transformer-like model, CRATE. Although designed with a principled objective, we identify an artifact in its forward construction and show that the simplest implementation can have the opposite effect in realizing its designed goal. We then provide implementation variants and investigate their layer-wise behaviors in optimizing SRR. An interesting finding is that alternative models exhibit similar behaviors, validating the use of SRR in designing Transformer-like models. Furthermore, we demonstrate its positive correlation to generalization and effectiveness over baselines. Driven by this connection, we show a simple way to use SRR as regularization to improve performance on CIFAR-10/100 datasets. Future direction may include applying layer-wise training and connecting SRR to the Forward-Forward algorithm [16], or exploring the impact of depth in the unrolled models.

## Limitations

This study has several limitations. Firstly, the conclusion that the SRR measure can be a strong indicator of generalization is limited to the CRATE family. Generalizing this conclusion to standard Transformers would be non-trivial, as the SRR measure is not properly defined when the query-key-value matrices have independent learnable parameters instead of shared ones. Secondly, the performance of a more faithful implementation (CRATE-N) falls behind the one with a simple manipulation (CRATE-T). This calls for a rigorous inspection of each component's functionality in the framework. Lastly, while we confirm the positive correlation to generalization, our analysis is limited in scale. Consequently, drawing definitive conclusions regarding whether SRR can be a principle or necessitates further engineering to push the model's limit is challenging. A Better and more systematic way is needed to determine whether SRR is principled for designing the Transformer-like models and quantify this relationship in an appropriate task, perhaps beyond classification.

## Acknowledgments

This work was supported in part by Natural Science Fund China (62306252), in part by the Hong Kong Research Grants Council General Research Fund (17203023), in part by the Hong Kong Research Grants Council Collaborative Research Fund (C5052-23G), in part by The Hong Kong Jockey Club Charities Trust under Grant 2022-0174, in part by the Startup Fund and the Seed Fund for Basic Research for New Staff from The University of Hong Kong, and in part by the funding from UBTECH Robotics.

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

# A   Complete Demonstrations of the Pitfalls

To give a clearer picture of how approximations affect the optimization of $R^c$, we provide the complete results with different update rules under the same simplified settings in the main text:

(a) Gradient descent on $R^c$.

(b) Gradient descent on the second-order Taylor expansion of $R^c$.

(c) Gradient descent on the first-order term of the Taylor expansion of $R^c$ (w/o second-order term).

(d) Gradient descent on the second-order term of the Taylor expansion of $R^c$ (w/o first-order term).

(e) Further adding softmax function upon (d).

The results in Figure 5 correspond to the above experiments. Gradient descent on $R^c$ did make it decrease across layers. Conversely, applying gradient descent on its second-order Taylor expansion resulted in an increase, indicating a potentially flawed approximation. Isolating gradient descent to the second-order term led to a rise in $R^c$, as opposed to the design purpose. Furthermore, incorporating the softmax function, a real-world operation examined in the main text, did not alter this conclusion.

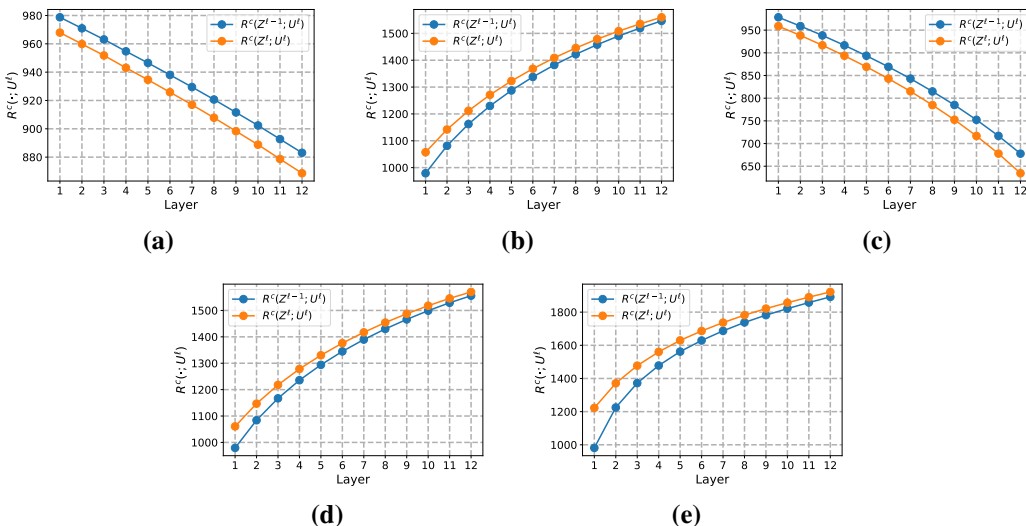

Figure 5: **(a)** Original gradient update, i.e, (6) **(b)** update from second-order Taylor expansion, i.e., (7) **(c)** update from removing the second-order term from (7) **(d)** update from removing the first-order term from (7) **(e)** update from further adding softmax, i.e., (8)

# B   Different Manipulations to the Output Matrix

As mentioned in Section 3, CRATE replaces the output matrix $U = [U_1, \ldots, U_K]$ in the MSSA operator with learnable $W$ (which is different from $U$). We then raise the following question on the manipulation of the output matrix: if we are free to adjust the output matrix while sacrificing interpretability, can we find more alternatives that can outperform CRATE-C or even CRATE? In practice, we have experimented with setting this matrix to an identity or fixed randomly initialized matrix, but only to discover that transpose performs best (Table 3). Therefore, CRATE-T is a feasible choice without introducing new parameters, which can be utilized to better understand the SRR principle and its connection to the performance.

We want to clarify that the analysis here intends to compare the variants with CRATE-C, not CRATE, because CRATE introduces learnable parameters that are less interpretable. We believe there are at least some interesting conclusions from the comparison: 1) CRATE-N achieves better performance by following the SRR principle more faithfully, shedding light on the connection of

Table 3: Top-1 accuracy for CRATE and its variants trained on CIFAR-10 from scratch (width $d = 384$).

| Models | CRATE-C | CRATE-N | CRATE-T | CRATE | CRATE-Fix | CRATE-Identity |
|---|---|---|---|---|---|---|
| # Params | 3.94M | 3.94M | 3.94M | 5.71M | 3.94M | 3.94M |
| Accuracy | 76.87 | 81.52 | 85.49 | 86.67 | 80.73 | 83.18 |

SRR to generalization; 2) We need to explore more design choices (e.g., CRATE-T, which may deviate from directly optimizing the SRR but still exhibit a similar architecture) to gain a complete understanding of the SRR principle for model performance (this motivates our Section 5).

## C   Experimental Details of Collecting Trained Models

Our experimental details to generate a family of trained models largely follow the previous work [18]. Models with heavy data augmentations tend to generalize better than those without them. It is therefore crucial to isolate the influence of data augmentations from the change of other hyperparameters. We choose to remove data augmentations during training to ensure that most models can be trained to meet the stopping criterion. We include Layer Normalization [2] before each operator during training, but also remove it when evaluating the complexity measures.

In this experiment, we vary across 5 sets of hyperparameters, i.e., batch size, initial learning rate, width, dropout probability, and model type. We present the choices of these hyperparameters in Table 4. Adam [21] is used as the default optimizer. Model depth is kept as $L = 12$ and number of subspaces $K = 6$. Dropout is applied after adding positional encoding, softmax function, and output projection in MSSA operator.

Table 4: Choices of hyperparameters.

| Hyperparameters | Choices |
|---|---|
| batch size | $\{64, 128\}$ |
| initial learning rate | $\{2 \times 10^{-5}, 1 \times 10^{-4}\}$ |
| width | $\{384, 768\}$ |
| dropout | $\{0.0, 0.1\}$ |
| model type | $\{$CRATE-C, CRATE-N, CRATE-T, CRATE$\}$ |

## D   Correlation of Complexity Measures when width $d = 768$

Table 5 and Figure 6 give results on correlation to generalization when width $d = 768$. We see that SRR is slightly better than other baseline measures in terms of overall $\tau$. In the axes of dropout and model type, however, it underperforms PAC-Bayes flatness measure. This implies that width could have a considerable influence on studying SRR as a complexity measure. We leave it for future work.

## E   Comparisons of Efficient Implementations

Table 6 compares different efficient implementations of SRR regularization. We find that randomly choosing layers to regularize generally worsens the performance. While regularizing shallower layers may bring more performance gain, leveraging the last layer already suffices to outperform the cross-entropy baseline. Specifying which layer to regularize could be expensive, especially when the model size grows. We opt for the last layer, which should be reasonable if depth scales. Our results indicate that this intuitive choice can already give consistent performance gains in different settings.

Table 5: Correlation of complexity measures with generalization gap (width $d = 768$).

| | Batch size | Learning rate | Dropout | Model type | Overall $\tau$ | $\Psi$ |
|---|---|---|---|---|---|---|
| $\ell_2$-norm | 0.000 | -0.375 | -0.625 | -0.250 | -0.310 | -0.313 |
| $\ell_2$-norm-init | 0.000 | -0.375 | -0.625 | -0.208 | -0.274 | -0.302 |
| # params | 0.000 | 0.000 | 0.000 | -0.295 | -0.188 | -0.074 |
| 1/margin | -0.125 | 0.375 | 0.625 | -0.208 | 0.173 | 0.167 |
| sum-of-spec | 0.000 | -0.375 | -0.625 | -0.375 | -0.310 | -0.344 |
| prod-of-spec | 0.000 | -0.375 | -0.625 | -0.417 | -0.339 | -0.354 |
| sum-of-spec/margin | 0.000 | -0.375 | -0.625 | -0.458 | -0.319 | -0.365 |
| prod-of-spec/margin | 0.000 | -0.375 | -0.625 | -0.417 | -0.327 | -0.354 |
| fro/spec | 0.000 | 0.375 | 0.500 | -0.083 | 0.242 | 0.239 |
| spec-init-main | 0.000 | -0.375 | -0.625 | -0.417 | -0.331 | -0.354 |
| spec-orig-main | 0.000 | -0.375 | -0.625 | -0.417 | -0.331 | -0.354 |
| sum-of-fro | 0.000 | -0.375 | -0.625 | -0.333 | -0.306 | -0.333 |
| prod-of-fro | 0.000 | -0.375 | -0.625 | -0.250 | -0.278 | -0.313 |
| sum-of-fro/margin | -0.125 | -0.375 | -0.500 | -0.167 | -0.286 | -0.292 |
| prod-of-fro/margin | -0.125 | -0.375 | -0.500 | -0.125 | -0.238 | -0.281 |
| fro-distance | 0.000 | -0.375 | -0.625 | -0.208 | -0.274 | -0.302 |
| spec-distance | 0.000 | -0.375 | -0.625 | -0.417 | -0.322 | -0.354 |
| param-norm | 0.000 | -0.375 | -0.625 | -0.250 | -0.310 | -0.316 |
| path-norm | -0.250 | -0.625 | 0.125 | -0.500 | -0.415 | -0.313 |
| pac-bayes-init | 0.000 | -0.375 | -0.625 | 0.250 | -0.214 | -0.188 |
| pac-bayes-orig | 0.000 | 0.375 | 0.625 | 0.167 | 0.315 | 0.292 |
| $1/\sigma$ pac-bayes-flatness | 0.000 | 0.375 | 0.688 | 0.573 | 0.337 | **0.409** |
| SRR | 0.125 | 0.500 | 0.250 | 0.375 | **0.407** | 0.313 |

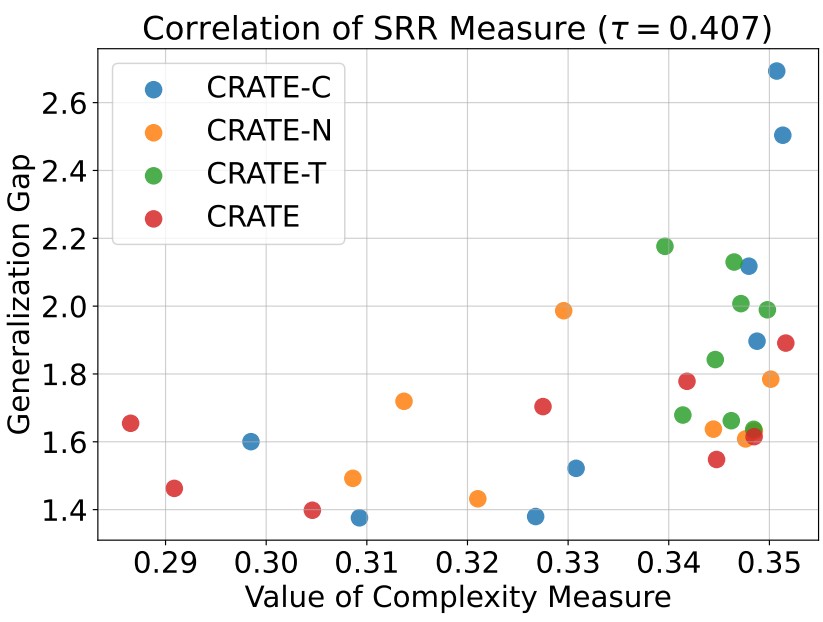

Figure 6: A scatter plot illustrating the value of SRR measure and generalization gap across CRATE and variants with network width $d = 768$.

Table 6: Top-1 accuracy for CRATE and its variants trained with efficient implementations of SRR regularization on CIFAR-10 from scratch (width $d = 384$).

| Training methods | CIFAR-10 | | | |
|---|---|---|---|---|
| | CRATE-C | CRATE-N | CRATE-T | CRATE |
| cross-entropy | 76.87 | 81.52 | 85.49 | 86.67 |
| + Layer 2 reg | 77.75 | **82.41** | **85.84** | **87.03** |
| + Layer 4 reg | **77.95** | 81.57 | 85.46 | 87.03 |
| + Layer 6 reg | 77.48 | 80.83 | 85.22 | 87.02 |
| + Layer 8 reg | 77.04 | 81.29 | 85.12 | 86.64 |
| + Layer 10 reg | 77.44 | 81.19 | 85.68 | 86.67 |
| + Layer 12 reg | 77.61 | 81.91 | 85.52 | 86.79 |
| + Random layer reg | 75.19 | 79.66 | 84.27 | 85.36 |

