# OpenReview forum: "An In-depth Investigation of Sparse Rate Reduction in Transformer-like Models"
_NeurIPS.cc/2024/Conference — NeurIPS 2024 poster_

### Official Review · Reviewer_7qne · 2024-07-14

**Soundness:** 3
**Presentation:** 3
**Contribution:** 2
**Rating:** 7
**Confidence:** 4

**Summary:**

The work investigates the design choices and analysis components in the recently proposed white-box transformer (CRATE). The work studies and analyzes the design rolling objective (SRR) in CRATE-like architecture.  The work also proposes using SRR as layerwise constraints and shows the SRR will further improve the model performance especially on the variants proposed in the paper. The work also investigates using SRR as a measure for generalization.

**Strengths:**

1. The work is well-written, with clear structures and language.
2. The work provides thorough studies into the design choices and justification of SRR as unrolling objectives.
3. The findings in the paper provide valuable additional insights to the original CRATE paper.
4. The analysis on generalization via Correlation of SSR is valuable that could potentially encourage future works.

**Weaknesses:**

1. The analysis in section 5 is very interesting. It will benefit the paper if the authors can provide more insights and investigation. For example, examining using SSR to measure other pretrained transformer-like architectures.
2. The analysis in Section 4 and experiments in Section 6 uses different alpha values. It could further improve the work if consistent alpha is used.

**Questions:**

NA

---

> ### Author Rebuttal · Authors · 2024-08-07
>
> **It will benefit the paper if the authors can provide more insights and investigation from section 5. For example, examining using SSR to measure other pretrained transformer-like architectures.**
>
> Evaluating $R^c(Z;U)=\sum_{k=1}^K \frac{1}{2}\log \operatorname{det}(I+\gamma (U_k^TZ)^T (U_k^TZ))$ in SRR need to specify $U$. However, it is not well defined on models other than CRATE and its variants like general transformers. Moreover, conducting correlation analysis needs to train a large number of models, but currently we lack the resources and time to accomplish it. Despite that, we will provide some preliminary results on evaluating SRR on a standard transformer in **Figure 5 in the pdf**. Specifically, we set $U=[U_1,\dots,U_K]$ equal to the query, key, and value projections respectively. This shows we may generalize the usage to a standard transformer, or even more general models. We will surely add this experiment and discussion in the revision.
>
> **The analysis in Section 4 and experiments in Section 6 uses different alpha values. It could further improve the work if consistent alpha is used.**
>
> We will change the Figure 1(a) using consistent $\alpha$ with that in Section 6 in the revision.

---

> > ### Comment · Reviewer_7qne · 2024-08-14
> > **Thank you for the rebuttal**
> >
> > I thank the author for the interesting rebuttal. The results of using SSR to measure other transformers are interesting. I encourage the author to add these experiments to the camera-ready and explore more.

---

### Official Review · Reviewer_SqrQ · 2024-07-15

**Soundness:** 3
**Presentation:** 3
**Contribution:** 3
**Rating:** 6
**Confidence:** 4

**Summary:**

The paper investigates a Transformer-like deep network architecture CRATE based on algorithmic unrolling of the sparse rare reduction objective function. It points out some pitfalls of the approximated layer operation where it does not decrease the R^c as it should, studies some alternatives, and also points out some surprising phenomena of training the unrolled architecture. Finally, it investiages whether SRR can be used as a regularizer in the loss function to promote generalization performance.

**Strengths:**

The investigation pointed out many interesting observations. For example,



1. the softmax attention update can be seen as an approximation to the lower-bound of R^c, an alternative view to the CRATE paper
2. Without Relu, the softmax attention update increases R^c as opposed to decreasing R^c, while with Relu added R^c decreases.
3. As the training proceeds, the SRR measure at any layer decrases as opposed to increases.
4. The SRR measure has the strongest correlation of the generalization gap (albeit i do not understand its definition, as illustrated in weakness), which motivated adding it to the learning objective as a regularizer for generalization.

**Weaknesses:**

Despite the interesting observations, there seem to be disconnections here and there.

## Section 4.1:



1. Lines 125 and 135-136 make conclusions based on how (6) is related to (7), but these conclusions are not justified.

    _It can be seen that this update step takes the gradient of a lower bound of R c (Z; U) and discards the first-order term._

* It is true that from (6) to (7) the first order term is removed as the authors claimed. Note however a softmax is also added.

    _When taking the gradient of R c to construct the MSSA operator, omitting its first-order term will produce a counterproductive effect._

* Maybe omitting the first-order term is not the (only) culprit, adding softmax could also play a role.
2. Line 141: _Therefore, one step of update (7) secretly maximizes R c_. I’m not sure about this either. Note that the green and orange curves are both quadratic functions, and are shifted/scaled versions of each other. Why would doing gradient descent with the orange be better than with the green?

    I would imagine at least two potential answers. 1) It depends on where gradient descent is initialized, i.e., at the left or right to the peak of the curve. But this needs to be justified better. 2) Using a quadratic approximation of the log of eigenvalues as in (6) is already problematic.

3. I think part of the issues above can be addressed by making a more complete experiment. More precisely, ​​using the same parameters settings as in making Figure 1(a), we should have curves resulting from
    1. the original gradient update
    2. the update from 2nd order Taylor expansion, i.e., equation (6)
    3. the update from removing the second order term from (6)
    4. the update from further adding softmax, i.e., (7)

## Section 4.2:



1. I do not understand the motivation of the variants proposed in the 4.2. ​​Granted, (7) is not the best approximation to the SRR. How do (8) and (9) make better approximations? It seems that they are just arbitrary modifications from (7) not serving any goals.

    If the argument, as the author states in lines 135-136, is that omitting the first-order terms in (6) is bad, then isn’t it natural to see what if one keeps the first-order terms?


## Section 5.6



1. Where is the generalization gap g(theta) defined? I do not seem to find them. I presume mu is any complexity measure, and tau is Kendall’s coefficient that is cited.
2. In table 2, the change of the accuracies by adding SRR regularization seems relatively small. I do not know if there is significance in it.

**Questions:**

See above

---

> ### Author Rebuttal · Authors · 2024-08-07
>
> **W1: Lines 125 and 135-136 make conclusions based on how (6) is related to (7), but these conclusions are not justified.**
>
> The derivations in (6-7) are rephrased from the original CRATE paper [1]. The softmax introduced after omitting the first-order term is quite intuitive, which converts auto-correlation to the distribution of membership of different subspaces. We refer the reviewer to Section 2.3 of [1] for discussion.
>
> **W1: Note however a softmax is also added from (6) to (7). Maybe omitting the first-order term is not the (only) culprit, adding softmax could also play a role.**
>
> Your concern that introducing softmax may also affect the optimization is understandable. But note that, as we mention in line 128, the update we use in the toy experiment is (7), which contains the softmax. Therefore, Figure 1(a) should be able to corroborate our analysis, and adding softmax does not change the conclusion.
>
> **W1: Therefore, one step of update (7) secretly maximizes $R^c$. I’m not sure about this either. Note that the green and orange curves are both quadratic functions, and are shifted/scaled versions of each other. Why would doing gradient descent with the orange be better than with the green?**
>
> The two answers you mention are reasonable. But we are not arguing which one is better. After all, the goal is to minimize the blue curve$-$the log function. If using the orange curve, we need to constrain the eigenvalues on the left of the peak, which is hard. The update (7) discards the first-order term, corresponding to the green curve. Now, minimizing the green curve makes the eigenvalues $(\ge 1)$ increase, hence maximizing the blue curve which corresponds to $R^c$. The quadratic function is the most straightforward approximation for designing the theory-grounded structure$-$MSSA operator. By optimizing other approximations of log functions, more powerful and more mathematically meaningful structures may emerge.
>
> **W1: part of the issues above can be addressed by making a more complete experiment**
>
> We follow these settings and provide the visualization in **Figure 4 in the pdf.** The results shows that the original gradient update of $R^c$ (a) with no approximation will monotonically decrease $R^c$, which is expected. Using the Taylor expansion approximation (b) will increase $R^c$, meaning that the eigenvalues $\gamma$ are mostly greater than one and the optimization happens to the right of the peak of the orange curve in Figure 1(b). Removing the first-order term of the Taylor expansion (d) monotonically increases $R^c$ which is the issue we specify in CRATE-C. Finally, further adding the softmax (e), leading to the update in (7), will not change the monotonic increase of $R^c$. We believe this figure will address most of the concerns in this part.
>
> **W2: I do not understand the motivation of the variants proposed in the 4.2. ​​Granted, (7) is not the best approximation to the SRR. How do (8) and (9) make better approximations?**
>
> These two variants have different motivations. CRATE-N is motivated to counteract the issue in CRATE-C where the update (7) increases the coding rate $R^c$, conflicting with the SRR principle. By moving in the opposite direction of CRATE-C, CRATE-N with (8) aims to implement the decrease in $R^c$ more faithfully, aligning better with the SRR principle. On the other hand, CRATE-T is not motivated for better alignment with the SRR principle. Instead, it is motivated by designing competitive alternatives of CRATE-C or even CRATE without introducing new parameters, since CRATE replaces the output matrix $U=[U_1,\dots,U_K]$ in CRATE-C with learnable $W$ (line 107) which sacrifices interpretability.
>
> **W2: If the argument, as the author states in lines 135-136, is that omitting the first-order terms in (6) is bad, then isn’t it natural to see what if one keeps the first-order terms?**
>
> Omitting the first-order term is bad when doing gradient *descent* because this will maximize $R^c$, but it is good when doing gradient *ascent* since this minimizes $R^c$ which is (8). It’s indeed natural to keep it, but we have to constrain the eigenvalues on the left to the peak of the orange curve, which is difficult.
>
> **W3: Where is the generalization gap $g(\theta)$ defined?**
>
> We measure the generalization gap as the difference between validation loss and training loss at convergence (training loss reaches 0.01), i.e., $L_{val}(w)- L_{train}(w)$. We can give a formal definition in the appendix in the revision.
>
> **W3: In table 2, the change of the accuracies by adding SRR regularization seems relatively small.**
>
> Our goal is not to demonstrate the superiority of SRR over other kinds of regularization or for substantial performance gains. Instead, we would like to validate and complement the conclusions in Section 5. SRR measure is shown to be better than sharpness as a predictor of generalization, then it should be reasonable to incorporate it into the training for improved generalization, similar to sharpness-aware minimization [2]. One direct approach is through regularization. We only provide preliminary results as a proof-of-concept. We believe there is room for engineering to make the results more significant.
>
> [1] Yu et al. White-Box Transformers via Sparse Rate Reduction. NeurIPS, 2023.
>
> [2] Foret, et al. Sharpness-aware Minimization for Efficiently Improving Generalization. ICLR, 2021.

---

> ### Comment · Reviewer_SqrQ · 2024-08-12
> **Thank you for the rebuttal**
>
> Thanks to the authors for their rebuttal!
>
> **Section 4.1/1**
>
> I agree that using softmax does not change the conclusion that update (7) is increasing the $R^c$ term (which is counterproductive) as shown in Figure (1).
>
> What I am not sure about is the apriori attribution of this counterproductive phenomenon to solely omitting first-order terms, as the paper writes on lines 135-136. Couldn't it be due to the softmax approximation? Do I miss something here?
>
> **Section 4.1/2**
>
> I see! So the thing that I was missing was the fact that $\lambda_i^k \geq 1 $ (which you mentioned on line 122), so one is always at the right half of the green curve. This could be highlighted better: e.g., you could make the left half of the green curve dashed, and/or reiterate the fact that $\lambda_i^k \geq 1 $ near line 140.
>
> **Section 4.1/3**
>
> Thanks for providing the experiments. They make the picture clear, and therefore I would suggest adding/referring to them in section 4.
>
> **Section 4.2**
>
> I think reviewers RBPW (W1), uGk9 (W1) and I share the same concern. It seems that CRATE-N and CRATE-T do not connect tightly with the findings in section 4.1.
>
> Perhaps the interesting finding is that of Table 2: Implementation-wise, one has to deviate from CRATE-C and use CRATE for maximized performance, and CRATE-N/-T serve as some sort of performance interpolations between CRATE-C and CRATE. It remains an open problem what the changes (from CRATE-C) do to the representation/optimization dynamics in terms of first principles.

---

> > ### Author Response · Authors · 2024-08-12
> >
> > We're glad that the additional experiments address your concerns in Section 4.1 and we will make corresponding improvements in the revision.  As for concerns in Section 4.2,  although CRATE outperforms CRATE-C, we believe there is no need to introduce new parameters to do so. Optimizing the SRR objective in the forward pass could matter, but reducing the SRR measure as a whole in (10) could be more important to correlate with improved generalization, as demonstrated in Section 5. CRATE-N/T are just instantiations of this possibility, though motivated differently. Indeed, we need to investigate further the role of ISTA block in (3) to have a deeper understanding of the framework.

---

### Official Review · Reviewer_uGk9 · 2024-07-15

**Soundness:** 2
**Presentation:** 3
**Contribution:** 2
**Rating:** 5
**Confidence:** 3

**Summary:**

This paper considers a recent line of transformer-like models called CRATE where each layer is designed to approximate a gradient-based optimization step of an information-theoretic objective function called sparse rate reduction. The contributions of the paper are: (1) investigating whether CRATE actually implements optimization of sparse rate reduction, and proposing variants of CRATE based on the observations, and (2) empirically analyzing the relation between the measured sparse rate reduction in CRATE (and the proposed variants) and their capabilities for generalization. The paper first shows that the approximation involved in the original derivation of CRATE results in cases where the forward pass of each layer does not implement gradient descent on sparse rate reduction, but does the opposite (ascent); from this observation, the authors propose CRATE-N and CRATE-T, where the former negates the output of a layer (before applying skip connection) and the latter transposes the output projection. Then, on CRATE and the proposed variants, the authors study how sparse rate reduction actually happens, and proceeds to studying the (causal) relationship between sparse rate reduction and generalization by measuring their correlation followed by showing that explicitly regularizing sparse rate reduction on the final layer leads to improved generalization.

**Strengths:**

- S1. The paper studies an interesting question of whether sparse rate reduction correlates with generalization in CRATE, among other questions studied in the paper. The proposed variants (CRATE-N and CRATE-T) are technically original as far as I am aware.
- S2. The results given in Table 1 that sparse rate reduction correlated well with generalization gap for models is interesting, in particular since it outperforms a sharpness-based measure which is known to correlate with generalization.
- S3. The paper is overall well written and easy to follow.

**Weaknesses:**

- W1. The motivations and technical soundness of the proposed variants of CRATE is unclear, especially given that they underperform on CIFAR-10/100 compared to the original CRATE as shown in Table 2. For CRATE-C, the authors point to Equation (7) and differentiate it from the original CRATE, but from the main text it seems Equations (5-7) describe the approximations involved in the original CRATE and how CRATE-C is different is not clear. Also, in Line 125, the authors state that "update step takes the gradient of a lower bound of $R^c(\mathbf{Z};\mathbf{U})$" which was a bit weird to me considering that the objective is minimizing $R^c(\mathbf{Z};\mathbf{U})$; I am not aware of optimization approaches that *minimize the lower bound* of the objective function, although this might be due to my limited knowledge in optimization literature. For CRATE-N, the authors claim in Lines 147-149 that "taking gradient ascent on the second-order term in the Taylor approximation of $R^c(\mathbf{Z};\mathbf{U})$ minimizes the eigenvalues and consequently $R^c(\mathbf{Z};\mathbf{U})$", which is not true for general objective functions as far as I am aware - please correct me if I am missing something. For CRATE-T, the authors motivate it in Line 152 by saying that "replacing the output matrix by learnable parameters is problematic ... it also breaks the inherent structures and sacrifices the mathematical interpretability", but then proceed to still use learnable output matrix, only transposing it as far as I understand. It is unclear to me what particular arguments the authors are trying to show with this variant.
- W2. In Section 4.3, the reported tendency of increasing sparse rate reduction measure after certain depth (Figure 2 and 3) does not agree with Figure 4 of the original CRATE paper [1], but proper discussion is not given. Also, although the authors have motivated CRATE-N (among other variants) from the pitfalls of the formulations of the original CRATE, it shows a similar tendency of worsening sparse rate reduction after certain depth. The experiments themselves are conducted on CIFAR-10/100, which is more limited compared to the similar analysis in Section 3.1 of the original CRATE paper [1] which has used ImageNet-1k.
- W3. In Section 6.2, in Line 271, it is not clear what the authors meant by "regularizing the training loss with sparse rate reduction for each layer ... lacks fine-grained controls over parameters". Regarding the results, the performance gain obtained by regularizing the last layer is marginal, and it is unclear if the gain is statistically significant, and whether it outweights the computational cost of computing the measure of sparse rate reduction. In fact, Table 5 shows that regularizing layer 4 achieves the best performance gain overall; the reason the authors have particularly chosen layer 12 to be shown in Table 2 is unclear. Lastly, Table 5 shows that random layer regularization using sparse rate reduction leads to degraded performances compared to not using regularization, and the reason is not clear: I believe underfitting is not likely as the dataset is CIFAR-10, which implies that random layer regularization using sparse rate reduction hurts generalization.

[1] Yu et al. White-Box Transformers via Sparse Rate Reduction (2023)

**Questions:**

- Q1. In Equation (12), how is the generalization gap $g(\theta)$ measured? Is it measured as the difference between the training loss (0.01) and test loss?

**Limitations:**

The authors have partially discussed the limitations in the checklist. I encourage the authors to create a separate section in Appendix.

---

> ### Author Rebuttal · Authors · 2024-08-07
>
> **W1:Unclear motivations/soundness of variants**
>
> CRATE-N aims to counteract issues in CRATE-C where the update can increase $R^c$, opposing SRR principle. By moving in the opposite way of CRATE-C, CRATE-N implements decrease in $R^c$ more faithfully, aligning better with SRR principle. CRATE-T addresses the discrepancy between CRATE and CRATE-C by questioning the replacement of the output matrix in MSSA operator with learnable parameters in CRATE. By exploring alternatives without new parameters, CRATE-T aims to enhance comprehension of SRR principle and improve performance of CRATE-C.
>
> **W1:Authors point to (7) and differentiate it from CRATE...how CRATE-C is different?**
>
> Sorry for making you confused. (5-7) describe the approximation and derivation of part of CRATE-C, and (7) is the update in CRATE-C. On the other hand, CRATE involves further modification-replacing matrix $U=[U_1,\dots,U_K]$ with learnable parameters $W$(line 106). The title of Section 4.1 should refers to "CRATE-C", not "CRATE".
>
> **W1:"...update step takes the gradient of a lower bound..." weird as the objective is minimizing $R^c$**
>
> You are right, minimizing the lower bound of the objective cannot reliably lead to the minimization of the original objective $R^c$. This is exactly the issue we want to highlight for CRATE-C.
>
> **W1:"taking gradient ascent...consequently $R^c$", not true for general objective functions**
>
> For the general function, you are correct. Yet, we're not using a general function. As in line 122, eigenvalues of a PSD matrix plus an identity matrix are definitely no less than one. When $\lambda \ge 1$, gradient ascent on the second-order term of $R^c$ (green curve in Figure 1(b)) will decrease $\lambda$, hence decreasing $R^c$ (blue curve). Therefore, our statement still holds.
>
> **W1:Authors motivate CRATE-T in "replacing output matrix...breaks the structures...", but then still use learnable output matrix, unclear arguments to show**
>
> The reason for sacrificing the interpretability is not about "learnable", it is about learnable parameters are "different" for output matrix and Q/K/V matrices. This can greatly enhance the performance. We then want to explore other designs not entirely derived from theory. For example, we have experimented with fixed weights and identity matrix in **Table 1 in the pdf**. We want to highlight designs different from $[U_1,\dots,U_K]$ in CRATE-C without new parameters.
>
> **W2:Increasing SRR after certain depth (Figure2/3), not agree with Figure4 of [1], need discussion.**
>
> There are several reasons: 1) Most importantly, different y axis. They plot the figures of compression and sparsity term separately, which fail to reflect the true tendency of compression term because they use $Y^\ell$ in (2). Instead, we visualize SRR measure as a whole and use $Z^\ell$ in (3); 2) They apply $\ell_2$ normalization before measurement of $R^c$ while we follow the definition. 3) CRATE-Base v.s. CRATE-Tiny configurations; 4) ImageNet-1k v.s. CIFAR10/100.
>
> **W2:Motivated from pitfalls of the original CRATE, CRATE-N still shows worsening SRR after certain depth.**
>
> SRR has two terms: $R^c(Z)$ and $\lambda \|Z\|_0-R(Z)$. Based on analysis in Section 4, the update in (8) can decrease $R^c$. Yet, the operation in (3) that aims to optimize the latter term also matters for the whole objective and may affect $R^c$ as well. This structure could have influences on the rise of the curves in Figure 2 and 3. How the ISTA block in (3) interacts with the MSSA block is an open question.
>
> **W2:Limited experiments compared to [1] that used ImageNet-1k.**
>
> We provide SRR measure similar to Figure 2 and 3 on ImageNet-1k in **Figure 2 in the pdf**. The models are trained with the official recipe [1]. We also provide the sparsity term in SRR measure $\|\boldsymbol{Z}\|_0/(d*N)$ in **Figure 3 in the pdf**. The sparsity term has a similar tendency with the SRR measure, and it goes up in the last few layers, which means that ISTA block also matters and need further research.
>
> **W3:Unclear what "lacks fine-grained controls over parameters" means.**
>
> What here emphasize the reg term in (10) is an average SRR measure over all layers, and is highly inefficient to compute. The SRR measure at some layer might be more important and should be optimized in isolation. Therefore we propose to regularize one specific layer at each iteration. We'll make this point clear in the revision.
>
> **W3:Reg on the last layer, marginal performance gain.**
>
> Our goal is NOT to demonstrate its superiority in performance gains. Instead, we want to complement the conclusions in Section 5. SRR measure is shown to be better than sharpness, then it should be reasonable to incorporate it into the training for generalization, similar to sharpness-aware minimization [2]. One direct approach is through regularization. We only provide preliminary results and leaves room for engineering to improve the results.
>
> **W3:Why layer 12 in Table2.**
>
> Specifying which layer to regularize could be computationally prohibitive, especially when the model size grows. We intuitively select the last layer, which should be reasonable if the depth of the models scales. Results in Table 2 indicate that this intuitive choice can already give consistent performance gains in different settings.
>
> **W3:Random layer reg in Table5 performs worse than w/o it, reasons unclear.**
>
> We also provide the accuracy on the training set of CIFAR10 in **Table 2 in the pdf**. The performance drops more on the training set than validation set. So underfitting could be the problem that hurts the generalization.
>
> **Q1:How is the generalization gap measured?**
>
> It's the difference between validation and training loss at convergence (training loss reaches 0.01).
>
> **Limitations.**  We will include the limitation section in the revision.
>
> [1] Yu et al. White-Box Transformers via Sparse Rate Reduction.
> [2] Foret, et al. Sharpness-aware Minimization for Efficiently Improving Generalization.

---

> > ### Comment · Reviewer_uGk9 · 2024-08-12
> >
> > Thank you for the comprehensive rebuttal, and I have read other reviews and responses as well. My original concerns were mostly addressed. I have adjusted my scores, but not very confidently since it seems a remaining limitation of the work is that CRATE-T, although motivated as a method that could outperform CRATE-C or CRATE by sacrificing interpretability (as in the response to W1 of reviewer RBPW), does not seem to outperform the original CRATE in the presented experimental results (e.g. in Table 2). This seems like a limitation as it means theoretical motivations were not precisely demonstrated in the experiments.

---

> > > ### Author Response · Authors · 2024-08-13
> > >
> > > Thank you for acknowledging our responses and raising the score, and we are happy to see that your concerns are mostly addressed. To clarify more, we want to explore architectures that may outperform CRATE-C, the conceptual framework, and hopefully be on par with CRATE without bringing in new parameters that are hard to analyze. Efforts are made both theoretically (CRATE-N) and empirically (CRATE-T). Yes, we also find that CRATE-T and other variants (Table 1 in the supplemented pdf) may not outperform CRATE, and this further motivates us to perform deeper analysis on the relationship between SRR measure and performance, i.e., Section 5, which reveals the positive effects of SRR. Therefore, we believe that SRR could be useful for enhancing performance, but its utilization or its guidance to the model architecture needs to be further investigated. In particular, it is possible to find a better approximation or implementation of SRR in building transformer-like models, that could achieve a better trade-off between interpretability and performance than the current CRATE. This will be left for our future exploration.

---

### Official Review · Reviewer_RBPW · 2024-07-16

**Soundness:** 2
**Presentation:** 4
**Contribution:** 3
**Rating:** 7
**Confidence:** 3

**Summary:**

This paper conducts an in-depth of study of CRATE, a previously proposed Transformer-like architecture to make deep learning more white-box. CRATE was motivated by sparse rate reduction (SRR), and it is a multi-layer architecture designed to optimize the SRR objective iteratively layer by layer. The authors first study whether the CRATE architecture can really optimize the SRR. They empirically show (in Figure 1) that the SRR objective actually increases with layer through a toy experiment, and the reason is that CRATE discards the first-order term in the Taylor expansion of the SRR objective. The authors then propose two variants, CRATE-N and CRATE-T, and study their behavior with experiments. Finally, the authors propose to use SRR as a complexity measure for predicting generalization. They experiment with a bunch of CRATE models and find that SRR correlates the best with the generalization performance. Motivated by this, the authors study whether SRR is a good regularizer.

**Strengths:**

This paper has clear strengths and clear weaknesses. Overall I enjoyed reading this paper, because
- It is very well written and easy to read. I hardly had any difficulty in understanding this paper.
- The analysis (especially in Section 4.1) is very clear. It clearly shows the problem of CRATE.
- The results are new as far as I know.
- The subject matter studied, which is a white-box deep learning architecture, is very important. I believe that many people will appreciate the analysis in this paper.

**Weaknesses:**

The paper has some clear weaknesses. I encourage the authors to revise this paper during the rebuttal period, and I'd be glad to provide my comments and feedback during revision.

- What is the point of proposing CRATE-N and CRATE-T? What problems do they solve? From the paper, I cannot understand why these two variants could be better than CRATE. Figures 2 and 3 don't show their superiority over original CRATE. Moreover, I don't see what interesting conclusions could be drawn from the comparison between these variants and the original CRATE.
- While the authors argue that CRATE can minimize the SRR objective, it seems to me from Figures 2 and 3 that this is not the case. The SRR at the last layer is still close to that at the first layer, especially for the two variants. Of course, SRR does decrease in intermediate layers, but the authors are not suggesting pruning the last layers of CRATE and using the intermediate representations.
- The experiment in Section 5 and Table 1 is confusing, and I am concerned whether it can be safely concluded from this experiment that SRR is a good proxy of generalization. First, I don't see why this is a "complexity measure". In statistical learning theory, a "complexity measure" is typically correlated with the size of a certain function class, but I don't see any function class here. Second, all models used in this experiment are CRATE models as shown in Table 3, which presumably optimize SRR. I am worried that the high correlation between SRR and generalization is a special property of CRATE models. To argue that SRR is a good proxy of generalization for all models, the authors ought to use models other than CRATE, that is models that have nothing to do with SRR.

**Questions:**

1. I cannot see why CRATE-N makes sense. If it is gradient ascent, how does it minimize $R^c$?
2. In line 170 the authors claimed that SRR "rises slightly in the subsequent layers". However, looking at Figures 2 and 3, my feeling is that it rises significantly. How do you define "slightly" here?
3. Why are you calling Eqn. (10) a "complexity measure"? Which function class is this complexity associated with? To me, Eqn. (10) is just the sum of coding rates at all layers.
4. Could you also use models that are not CRATE in the experiment of Section 5, and see if SRR can still predict their generalization?

**Summary:** I really like the analysis in Section 4.1 and think that this is a great contribution. However, I also have some big concerns with the subsequent sections. Currently I am rating this paper "borderline accept". I encourage the authors to revise this paper during rebuttal. Based on the final version, I could upgrade my rating to "accept".

**Limitations:**

Limitations are not discussed. I encourage the authors to add a limitation section in the revision.

**Post-rebuttal note:** After discussing with my fellow reviewers and the AC, I raised my rating to 7.

---

> ### Author Rebuttal · Authors · 2024-08-07
>
> **W1: What is the point of proposing CRATE-N and CRATE-T? What problems do they solve?**
>
> The goal of developing CRATE-N is to address the potential issue of CRATE-C. As we pointed out that the update (7) in CRATE-C, which performs a gradient descent only for the second-order term, could maximizing $R^c$, rather than minimizing it. This contradicts the principle of SRR. Therefore, CREATE-N is designed by seeking the opposite direction of CRATE-C, which could perform decreases of $R^c$ more faithfully, thus aligning better with the SRR principle.
>
> The goal of developing CRATE-T is to address the misalignment between CRATE and CRATE-C (i.e., the misalignment between CRATE and the SRR principle). As mentioned in line 107, CRATE replaces the output matrix $U=[U_1,\dots,U_K]$ in the MSSA operator with learnable $W$ (which is different from $U$). We then raise the following question on the manipulation of the output matrix: if we are free to adjust the output matrix while sacrificing interpretability, can we find more alternatives that can outperform CRATE-C or even CRATE? Therefore, CRATE-T is a feasible choice without introducing new parameters, which can be utilized to better understand the SRR principle and its connection to the performance. We will revise our Section 4 accordingly to improve clarity.
>
> **W1: Conclusions drawn from the comparison with CRATE**
>
> We want to clarify that our analysis intends to compare the variants with CRATE-C, not CRATE, because CRATE introduces learnable parameters $W$ that are less interpretable. We believe there are at least some interesting conclusions from the comparison: 1) CRATE-N achieves better performance by following the SRR principle more faithfully, shedding light on the connection of SRR to generalization; 2) We need to explore more design choices (e.g., CRATE-T, which may deviate from directly optimizing the SRR but still exhibit a similar architecture) to gain a complete understanding of the SRR principle for model performance (this motivates our Section 5).
>
> **W2: SRR at the last layer close to that at the first layer, especially for the two variants. Authors not suggesting pruning the last layers of CRATE and using the intermediate representations.**
>
> The SRR objective contains minimizing two terms: $R^c(Z)$ and $\lambda\|Z\|_0-R(Z)$. Based to analysis in Section 4, the update in (8) can decrease $R^c$. Yet, the operation in (3), i.e., ISTA operation, designed to optimize the latter term also matters for the whole objective. This structure may have effects on the rise of curves in Figure 2 and 3. In fact, the sparsity term $\|Z\|_0$ has been discovered to shoot up in the last layer in Figure 3 of the original CRATE paper [1]. We also make similar discoveries in other variants (see **Figure 3 in the pdf**).
>
> To see if this rise affects the intermediate presentations, we present the result of linear probing on them in **Figure 1 in the pdf**. It shows that the representations becomes more linearly separable as the layer goes deeper, although SRR is not well-optimized in the last few layers. This suggests there might be a trade-off between SRR, presumably sparsity, and representation learning in the last few layers. This also means SRR does not necessarily indicate linear separability, especially when the architecture does not faithfully implement the SRR objective.
>
> **W3 & Q3: Why is SRR a "complexity measure". Don't see any function class here.**
>
> Here we use "complexity measure" as we think higher SRR measure implies more complicated models, then it can be viewed as a "complexity measure" for a model. We agree that the rigorous definition of it should be for a certain function class. We will replace it with "proxy/measure of generalization" in the revision.
>
> **W3 & Q4: Worried that the high correlation between SRR and generalization is a special property of CRATE models. Could you also use models that are not CRATE in the experiment of Section 5, and see if SRR can still predict their generalization?**
>
> This is a good question. This is also one of the motivations to consider CRATE-T, which is not intentionally designed to minimize or maximize SRR. This variant can expand the candidate classes of transformer-like models in the correlation analysis. We will use more models such as those replacing the output matrix with randomly initialized and fixed weights or just identity matrix. We will also include more general models, such as standard transformers and CNNs for the experiment in Section 5. We have performed the preliminary analysis for transformers (see **Figure 5 in the pdf**). However, since the correlation analysis requires training a large number of models, we are short of time and computing resources to complete during the rebuttal phase. We will definitely include the experiments and discussions in the revision.
>
> **Q1: Cannot see why CRATE-N makes sense. If it is gradient ascent, how does it minimize $R^c$?**
>
> CRATE-N actually performs gradient ascent on the second-order term in (5). This corresponds to the green curve in Figure 1(b), which tends to make the eigenvalue $\lambda$ smaller (as the optimization is performed for $\lambda\ge1$ and small $\lambda$ increases the green curve). On the other hand, the right-hand side of (5) itself, which can be understood as the orange curve in Figure 1(b), has an opposite pattern as the green curve, which will decrease when $\lambda$ becomes smaller enough. This leads to the minimization of $R^c$ (which can be expressed with a logarithmic function of eigenvalue $\lambda$).
>
> **Q2: In line 170 the authors claimed that SRR "...rises slightly...". But it seems to me that it rises significantly.**
>
> We will make our description more accurate in the revision.
>
> **Add a limitation section in the revision.** We will add a limitation section in the appendix in the revision.
>
> [1] Yu et al. White-Box Transformers via Sparse Rate Reduction. NeurIPS, 2023.

---

> > ### Comment · Reviewer_RBPW · 2024-08-09
> > **Response**
> >
> > I thank the authors for the rebuttal. I like the additional experimental results presented in Figures 1 and 5 of the new pdf, and I encourage the authors to do more experiments even after the rebuttal period and include the new results in the new version. Most of my questions have been addressed, and I am inclined to accept this paper. I will discuss with my fellow reviewers and the AC and notify the authors if I raise my rating.

---

> > > ### Author Response · Authors · 2024-08-10
> > >
> > > We are glad that our response helps address your questions. Thank you for recognizing the additional experiments, and we will include these and more in the revision.

---

### Author Rebuttal · Authors · 2024-08-07

We sincerely appreciate the thoughtful reviews and comments provided by all reviewers. Below, we address the main points raised, details can be found in corresponding blocks for each reviewer:

- Reviewer RBPW questioned the role of different variants and the behavior of the SRR objective. We clarify that the variants have different motivations and highlight the benefits of adhering closely to the SRR objective and its relation to intermediate representations in the linear probing experiment.

- Reviewer uGk9 primarily raised confusion and concerns on the specification of problems and called for the use of complex datasets. We address the concerns regarding the performance in the experiment and elucidate the potential reasons for particular phenomena with the help of a larger dataset. Clearer descriptions are given to explain the details on the analysis.

- Reviewer SqrQ suspected the effects of approximation on the design goal and pointed out the missing definition and discussions. We provide similar toy experiments with different settings to illuminate the problems and make clear the interpretation regarding the figures.

- Reviewer 7qne acknowledged the significance of the experimental results and provided suggestions to improve the work.

---

### Decision · Program_Chairs · 2024-09-25

**Decision:**

Accept (poster)

**Comment:**

The authors conduct an in-depth investigation of CRATE, a previously proposed Transformer-like architecture derived from unrolled optimization of an information-theoretic objective function defined on representations referred to as “sparse rate reduction” (SRR), with subsequent approximation. The authors present a careful analysis of CRATE, showing in a toy experiment that these approximations (made for computational efficiency and performance reasons) actually lead the network to increase the SRR from layer-to-layer, rather than decrease it as per the design, and proposing architectural variants (“CRATE-N” and “CRATE-T”) designed to correct this discrepancy without sacrificing downstream performance. To better understand the remaining gap between CRATE’s performance and the performance of the proposed variants, they propose to use SRR as a complexity measure for predicting generalization; experiments on several CRATE model variants and transformers (post-rebuttal) highlight that SRR correlates the best among different generalization measures (e.g., sharpness) with the generalization performance.

After the rebuttal, in which authors clarified many vague points and presented new experiments corroborating their arguments on the relationship between SRR and generalization on non-CRATE architectures, scores for the paper increased, converging on accept. Reviewers appreciate the careful design of the paper’s experimental and theoretical studies and the additional insight it yields towards understanding generalization in transformer-like architectures. The AC concurs with the reviewers’ judgment, and recommends acceptance. The authors are encouraged to incorporate all feedback from the reviewers into the revision, especially regarding additional promised experiments on non-CRATE variants.